# Targeted Therapies in Colorectal Cancer: Recent Advances in Biomarkers, Landmark Trials, and Future Perspectives

**DOI:** 10.3390/cancers15113023

**Published:** 2023-06-01

**Authors:** Joao Manzi, Camilla O. Hoff, Raphaella Ferreira, Agustin Pimentel, Jashodeep Datta, Alan S. Livingstone, Rodrigo Vianna, Phillipe Abreu

**Affiliations:** 1Miami Transplant Institute, Jackson Memorial Hospital, University of Miami, Miami, FL 33136, USA; joao.manzi@fm.usp.br (J.M.); camilla.hoff@fm.usp.br (C.O.H.); raphaelladpferreira@gmail.com (R.F.); r.vianna@med.miami.edu (R.V.); 2Sylvester Comprehensive Cancer Center, University of Miami, Miami, FL 33136, USA; apimentel@med.miami.edu (A.P.); jash.datta@med.miami.edu (J.D.); alivings@med.miami.edu (A.S.L.)

**Keywords:** colorectal cancer, biomarkers, liver transplant, oncology, surgical oncology, colorectal liver metastasis

## Abstract

**Simple Summary:**

Colorectal cancer is one of the leading causes of cancer-associated deaths worldwide. From the 1960s to 2002, the standard systemic treatment consisted of cytotoxic chemotherapy. However, in the past 20 years, the development of targeted therapies has revolutionized care for CRC. Dozens of new drugs have been developed specifically targeting actionable and identifiable molecular biomarkers expressed by these tumors. This growing field, precision oncology, has been responsible for a dramatic drop of more than 50% in the CRC mortality rate in the United States. As research into this field exponentially expands, new targets are being uncovered, and new drugs are being developed. This review aims to summarize the recent advances in biomarkers, landmark trials, and future perspectives for colorectal cancer.

**Abstract:**

In 2022, approximately 600,000 cancer deaths were expected; more than 50,000 of those deaths would be from colorectal cancer (CRC). The CRC mortality rate in the US has decreased in recent decades, with a 51% drop between 1976 and 2014. This drop is attributed, in part, to the tremendous therapeutic improvements, especially after the 2000s, in addition to increased social awareness regarding risk factors and diagnostic improvement. Five-fluorouracil, irinotecan, capecitabine, and later oxaliplatin were the mainstays of mCRC treatment from the 1960s to 2002. Since then, more than a dozen drugs have been approved for the disease, betting on a new chapter in medicine, precision oncology, which uses patient and tumor characteristics to guide the therapeutic choice. Thus, this review will summarize the current literature on targeted therapies, highlighting the molecular biomarkers involved and their pathways.

## 1. Introduction

Cancer is the second leading cause of death in the United States of America (USA) today, second only to heart disease [1]. In 2022, approximately 600,000 cancer deaths were expected; more than 50,000 of those deaths would be from colorectal cancer (CRC) [1]. Despite a recent increase in incidence in the younger population, which is not yet subjected to routine CRC screening [2], the CRC mortality rate in the US has decreased in recent decades, with a 51% drop between 1976 and 2014 [3]. This drop is attributed, in part, to the tremendous therapeutic improvements, especially after the 2000s, in addition to increased social awareness regarding risk factors and diagnostic improvement [3]. For patients who develop metastatic CRC (mCRC), approximately 50–60% of all patients with CRC [4], the standard treatment is antineoplastic agents aimed at improving and prolonging the quality of life [5].

Five-fluorouracil, irinotecan, capecitabine, and later oxaliplatin were the mainstays of mCRC treatment from the 1960s to 2002 [5]. Since then, more than a dozen drugs have been approved for the disease, betting on a new chapter in medicine, precision oncology, which uses patient and tumor characteristics to guide the therapeutic choice [5]. More than half of the patients with mCRC have tumors with specific molecular profiles for which targeted therapies or immunotherapies are already available [6].

In this context of intense innovation, which is changing clinical practice and patient outcomes, understanding the mechanisms underlying these new medications has become imperative in medical practice [7]. Thus, this review will summarize the current literature on targeted therapies, highlighting the molecular biomarkers involved and their pathways.

## 2. Biomarkers and Targeted Therapies

### 2.1. Vascular Endothelial Growth Factor (VEGF)

The VEGF system is one of the most relevant and researched factors of angiogenesis, the formation, and the organization of new vessel arrangements, a critical step in tissue growth [8,9]. Hypoxia, oncogenes, and various cytokines may promote its transcription, increasing its action on the tissues in which it occurs [10]. In the neoplastic microenvironment, constant hypoxia promotes the tumor endothelium to continuously release VEGF in an autocrine manner, promoting cell growth and support [11,12,13].

VEGF can be used as a biomarker for metastatic colorectal cancer (mCRC), helping to understand and treat patients better [14]. In a retrospective analysis, Jurgensmeier et al. discussed “Cediranib with fluorouracil, leucovorin, and oxaliplatin (mFOLFOX6) versus bevacizumab with mFOLFOX6 as first-line treatment for patients with advanced colorectal cancer” (HORIZON III), a randomized controlled trial, concluding that high baseline VEGF values are associated with worse Progression-Free Survival (PFS) and Overall Survival (OS) in mCRC patients, independent of treatment [15]. A correlation was also observed between VEGF expression and the development of metastatic disease in patients who did not receive adjuvant chemotherapy when analyzed over a 5-year period [16].

The VEGF family contains VEGF-A, VEGF-B, VEGF-C, VEGF-D, and Placental Growth Factor, and it is the primary agent in the regulation of angiogenesis, promoting the maintenance, proliferation, migration, and differentiation of endothelial cells [17]. It is possible to differentiate VEGF-A more precisely since: multiple isoforms are created from a single gene, generating variants with different characteristics and molecular weights [18].

At least 12 isoforms are formed: VEGF-A111, VEGF-A121, VEGF-A121b, VEGF-A145, VEGF-A148, VEGF-A162, VEGF-A165, VEGF-A165b, VEGF-A183, VEGF-A189, VEGF-A189b, and VEGF-A206 [19,20]. This more precise classification is relevant because each subtype reflects different characteristics [17,20]. The most common pro-angiogenic form is VEGF-A165a, while the most common anti-angiogenic form is VEGF-A165b, pointing to very distinct types of behavior [17,21]. The VEGF-A189 isoform is correlated with a higher rate of liver metastasis and a worse prognosis [22]. There is also evidence that VEGF-A121, which can be detected in blood, may be a good predictor of a patient’s response to anti-angiogenic treatment [23].

The strong correlation between VEGF as an angiogenic factor and the development and progression of colorectal cancer leads to its usability for therapeutic purposes. Currently, the principal medications that act on this system are bevacizumab, Ziv-Aflibercept, Ramucirumab, and Regorafenib [24].

Bevacizumab, a humanized monoclonal antibody that binds to VEGF-A isoforms, is the first-line anti-angiogenic treatment for advanced or metastatic CRC [25]. It is usually used in combination with oxaliplatin-based therapies [25]. Its antitumor action can be described in three ways: 1—A decrease in the number of vessels, either by inhibiting the formation of new ones or by regressing newly formed vessels; 2—Changing the functionality of these vessels; 3—Direct effects on the tumor [25,26,27,28].

A recent meta-analysis from Baraniskin et al. indicates that the use of bevacizumab together with 5-FU-containing therapies significantly increases PFS (Hazard Ratio [HR], 0.71; 95% CI, 0.65–0.77; *p* < 0.00001) and OS (HR, 0.85; 95% CI, 0.78–0.94; *p* = 0.0008) [29]. When the analysis was restricted to fluoropyrimidine monotherapy, the addition of bevacizumab led to a significant increase in PFS (HR, 0.57; 95% CI, 0.48–0.66; *p* < 0.00001) and OS (HR, 0.83; 95% CI, 0.70–0.98; *p* = 0.03) [29].

Ziv-aflibercept is a recombinant protein composed of extracellular parts of the vascular endothelial growth factor receptor 1 and 2 (VEGFR-1 and VEGFR-2), linked to the constant fragment (Fc) region of human immunoglobulin G 1 (IgG1) [30]. It acts as a VEGF receptor decoy, binding to VEGF factors, thus preventing their efficient action [31]. Its role is still less defined than that of bevacizumab, especially when it comes to first-line treatment, but apparently it presents itself as a good tool for patients who, despite treatment with bevacizumab, continue with disease progression [32].

In the “VEGF trap (aflibercept) with irinotecan in colorectal cancer after failure of oxaliplatin regimen trial” (VELOUR trial), the combination of Ziv-Aflibercept with infusional fluorouracil, leucovorin, and irinotecan (FOLFIRI) was studied in patients with mCRC, previously treated with oxaliplatin, including those previously treated with bevacizumab [33]. A significant improvement was observed in the OS compared with that with FOLFIRI plus placebo (HR, 0.817; 95% CI, 0.713–0.937; *p* = 0.0032), with an increase in the median OS to 13.5 months using aflibercept plus FOLFIRI compared with 12.06 months with the use of placebo plus FOLFIRI [33]. There was also a significant increase in PFS, from 4.67 months to 6.9 months [34]. The response rate was 19.8% (95% CI, 16.4–23.2%) with aflibercept plus FOLFIRI versus 11.1% (95% CI, 8.5–13.8%) using placebo plus FOLFIRI (*p* = 0.0001) [33].

Ramucirumab is a complete human IgG-1 monoclonal antibody, which binds with high affinity to the extracellular domain of VEGFR-2, disrupting the VEGF-mediated signaling pathway and promoting an antitumor effect [35,36]. The randomized, double-blind, multicenter, phase III “Ramucirumab versus placebo in combination with second-line FOLFIRI in patients with metastatic colorectal carcinoma that progressed during or after first-line therapy with bevacizumab, oxaliplatin, and fluoropyrimidine trial” (RAISE trial) evaluated the use of this medication combined with FOLFIRI as a second-line versus placebo plus FOLFIRI in patients with mCRC [37]. After a follow-up of 21.7 months, the median OS was 13.3 months (95% CI 12.4–14.5) using Ramucirumab plus FOLFIRI, while it was 11.7 months (95% CI, 10.8–12.7) for the placebo group (HR 0.844; 95% CI 0.730–0.976; *p* = 0.0219). The PFS was 5.7 months (95% CI, 5.5–6.2) in the first group versus 4.5 months (95% CI 4.2–5.4) in the placebo group (HR 0.793; 95% CI-0.697–0.903; *p* = 0.0005) [37,38].

Regorafenib is a multiple kinase inhibitor already approved for the treatment of patients with CRC refractory to other therapies worldwide [39]. Originally, Regorafenib was developed as a V-raf-1 murine leukemia viral oncogene homolog 1 (RAF1) inhibitor, but in preclinical experiments, its usefulness as a multi-targeting kinase inhibitor with a wide therapeutic range was observed [40,41]. In the multicenter, randomized, placebo-controlled, “Regorafenib monotherapy for previously treated metastatic colorectal cancer” (CORRECT) study, Grothey et al. evaluated the use of Regorafenib as monotherapy in patients with mCRC previously treated with standard therapies [42]. In this study, the median OS was 6.4 months in the Regorafenib group compared with 5 months in the placebo group (HR 0.77; 95% CI 0.64–0.94; *p* = 0·0052) [42]. The “Regorafenib for patients with metastatic colorectal cancer who progressed after standard therapy” study (CONSIGN study) confirmed these results and found an overall median PFS of 2.7 months (95% CI), 2.5 months for KRAS mutant tumors, and 2.8 months for KRAS wild-type tumors [42].

### 2.2. Epidermal Growth Factor Receptor (EGFR) Pathway

EGFR belongs to the Erythroblastosis Oncogene B/Human Epidermal Growth Factor Receptor (HER) family and has four members: erythroblastic leukemia viral oncogene homolog 1 (ErbB1) (EGFR/HER1), ErbB2 (HER2), ErbB3 (HER3), and ErbB4 (HER4) [43]. Once activated, this receptor triggers a series of downstream intracellular signaling pathways, notably RAS/RAF/Mitogen-activated protein kinase (MEK)/extracellular-signal-regulated kinase (ERK); Phosphatidylinositol 3-kinases/AKT (PI3K/AKT); and Janus kinase-signal transducer and activator of transcription 3 (JAK/STAT3), which regulate the growth, migration, invasion, and survival of these cells [44,45]. The hyperactivation and hyperexpression of receptors in this family are related to several cancers, such as head and neck, stomach, colorectal, prostate, pancreas, and lung [43], and it may indicate a worse prognosis, survival, resistance to specific drugs, and higher metastasis rate [45,46,47].

Over-expression of EGFR is found in 65–75% of patients with advanced colorectal cancer [48]. Epidermal Growth Factor (EGF) and EGFR levels are significantly higher in malignant zones of colorectal cancer specimens than in other nearby regions [49]. The presence of high levels of EGFR in CRC justifies the use of targeted therapies for this pathway [50]. Cetuximab and panitumumab are the most used treatments that fit this category for CRC [24].

Cetuximab is an IgG antibody that, after competitive binding to the external domain of the receptor, promotes the internalization and destruction of EGFR [51], resulting in an inhibition of cell growth, a decrease in the production of matrix metalloproteinase (MMP) and VEGF, and induction of apoptosis [49,52,53].

Cetuximab, in conjunction with other chemotherapies, improves progression control, as noted by the Phase III “Irinotecan and 5-FU/FA with or without cetuximab in the first-line treatment of patients with mCRC” (CRYSTAL) trial, which demonstrated superior median progression-free survival times with FOLFIRI plus cetuximab compared with FOLFIRI alone (8.9 vs. 8 months, HR 0.85; 95% CI), but with an OS without significant differences (HR 9.93; 95% CI) [54]. The use of only cetuximab after the combined use of FOLFOX plus cetuximab maintained a similarly elevated PFS, but with fewer adverse effects [54].

Skin toxicity with cetuximab has an 80% of frequency, infusion-related reactions are present in 90% of the patients in the first infusion, and 48% for anemia [50]. Panitumumab, a fully humanized antibody that was developed, does not trigger antibody-dependent cell-mediated cytotoxicity, granting a different profile of toxicity with lower rates of anemia (48% vs. 5.2%; *p* < 0.001) and infections (22,4% vs. 12.5%, 95% CI, *p* < 0.001), but with higher rates of discontinuation (13.5% vs. 6.9%, 95% CI, *p* < 0.001) and fatal serious adverse events (SAE) (2.9% vs. 4.5%, 95% CI, *p* = 0.004) [55,56].

The “Panitumumab with infusional fluorouracil, leucovorin, and oxaliplatin (FOLFOX4) versus FOLFOX4 alone as first-line treatments in patients with previously untreated mCRC” (PRIME) trial compared panitumumab in conjunction with FOLFOX versus FOLFOX alone, finding better PFS in the panitumumab group (10 vs. 8.6 months, 95% CI 0.66–0.97, HR 0.80, *p* = 0.01), as well as better OS (23.9 vs. 19.7 months, 95% CI 0.67–1.02, HR 0.83, *p* = 0.17) [57,58]. The “Panitumumab Efficacy in combination with mFOLFOX6 against bevacizumab plus mFOLFOX in mCRC subjects with KRAS WT tumors” (PEAK) study found that median PFS was longer in the panitumumab group in relation to the bevacizumab group (12.8 vs. 10.1 months, HR = 0.68, 95% CI), and the median OS was 36.9 vs. 28.9 months, HR = 0.76, 95% CI [59]. Panitumumab has been shown to have a higher proportion of patients achieving early tumor shrinkage (ETS) in left-sided disease patients than in right-sided disease patients [60].

Both drugs, cetuximab and panitumumab, are considered first-line treatments for CRC [23] with no inferiority between them, as well as side effects, as shown in the phase III Panitumumab versus cetuximab in patients with chemotherapy-refractory wild-type KRAS exon 2 mCRC (ASPECCT) study, with an OS of 10.0 months for cetuximab and 10.4 months for panitumumab, 95% CI 0.839–1.113, HR 0.97, *p* < 0.0007 for noninferiority [61].

#### 2.2.1. EGFR-Related Resistance Mechanism

Treatment using Anti-EGFR monoclonal antibodies (MoAb), such as cetuximab or panitumumab, is only effective in a portion of patients due to, among others, non-expression of EGFR and mutations, both in EGFR itself and at other points in the pathway, which can lead to the development of resistance [62,63,64,65].

Mutations directly in the EGFR or a low expression of EGFR can decrease the benefit of anti-EGFR therapies [66]. The most common mutation in patients with CRC is the substitution of serine to arginine at amino acid 492 (S492R) in the extracellular domain of EGFR, which causes a decrease in response to cetuximab and panitumumab [67]. The incidence of EGFR mutation varies across studies by region. In Greece, patients with non-Small-cell Lung Carcinoma (NSCLC) had an EGFR mutation rate of 28% [68], while in South Korea, the rate was 15% [69]. In colorectal cancer patients, the incidence in Western countries of EGFR mutation was 0.34% [70], while it was 12% in Japan and 22.4% in South Korea [71,72]. Patients from the Middle Eastern population with colorectal cancer had an EGFR mutation rate of 0% [73].

EGFR-targeted therapies, such as cetuximab and panitumumab, may also affect neighboring stromal cells near the cancer cells, promoting an increase in the secretion of EGF, inducing resistance against EGFR-targeted therapies through continuous mitogen-activated protein kinase (MAPK) signaling [65].

#### 2.2.2. Resistance Mechanism—RAS Family

The RAS family, composed mainly of Kirsten rat sarcoma viral oncogene homolog (KRAS), neuroblastoma RAS viral oncogenes homolog (NRAS), and Harvey rat sarcoma viral oncogene homolog (HRAS), participates in the EGFR pathway, acting in the transducing and auto-inactivation of this pathway [66]. Mutations in this family are considered a major cause of resistance to anti-EGFR therapy [74,75,76,77]. The incidence of RAS mutations in CRC is about 53% [78], mainly in codons 12 or 13 of exon 2 (81–96% of all KRAS mutations), and 4–19% at exon 3 or exon 4 [66,79,80]. Therefore, the RAS genotype has become an essential factor in the therapeutic decision of patients with mCRC [48].

RAS inhibitors have recently been studied to overcome this obstacle [81]. Among the studies of RAS-targeted therapies, sotorasib, which acts selectively and irreversibly on KRAS-mutated glycine 12 to cysteine (G12C), is an example of a medication that promoted better responses to anti-EGFR in patients with RAS mutations, which is why it was approved by the Food and Drug Administration (FDA) in May 2021, although having no effect in patients with mutations at codons 12, 13, and 61 [82,83]. However, acquired resistance is anticipated, limiting its long-term clinical use due to secondary mutations causing resistance in sotorasib in vitro, and requiring a switch to two other drugs, BI-3406 plus Trametinib, to overcome this new obstacle [84]. Medications targeting other mutations, such as dasatinib, showed results in vitro but did not reproduce results in Phase IB/II studies of 77 patients with previously treated mCRC [85]. Some non-targeted drugs showed positive results in these patients [86]. Some studies indicate that simvastatin plus cetuximab reduces the proliferation of KRAS mutant cells and suppresses v-raf murine sarcoma viral oncogene homolog B1 (BRAF) activity in these patients [87,88], and that Metformin reverses acquired resistance in KRAS to anti-EGFR drugs [89].

#### 2.2.3. Resistance Mechanism PI3KCA/AKT

The PI3K/AKT signaling pathway, one of the components of the EGFR pathway, when activated, is an essential contributor to the growth, proliferation, and survival of multiple solid tumors [90,91]. The most frequently mutated gene that enhances this pathway is Phosphatidylinositol-4,5-bisphosphate 3-kinase, catalytic subunit alpha (PIK3CA), mainly in exon 9, 20, or both [92,93]. The frequency of these mutations in patients with CRC changes according to the detection technique used [94], but varies between 7 and 32% [93,94,95]. There is a positive association between PIK3CA mutations and clinical features, such as proximal location (OR = 1.79; 95% CI: 1.39–2.29) and mucinous histology (OR = 1.86; 95% CI: 1.50–2.31) [94]. It is also related to a worse prognosis [93,96].

PI3K inhibitors may contribute to the control of tumor growth, regardless of driver genotypes [91]. Mutations in PIK3CA lead to the permanent activation of the EGFR pathway, promoting resistance to EGFR-blocking effects and decreasing the response rate of anti-EGFR therapies, with a relative risk of 0.56 (0.38–0.82, 95% CI) [97]. Other drugs on the PI3K axis are being developed to act in cases of PI3KCA mutations, such as AKT inhibitors, mammalian target of rapamycin (mTOR) inhibitors, and dual inhibitors of PI3K and mTOR [93], but they are still in the initial phases [98].

In patients with mutated PI3KCA colorectal cancer, regular use of aspirin is associated with a blockade of the PI3K pathway, causing suppression of cancer cell growth and induction of apoptosis [99,100]. The post-diagnosis use is associated with a higher OS (HR = 0.83; 95% CI, 0.75–0.9) and superior colorectal cancer-specific survival (HR = 0.78, 95% CI, 0.66–0.92) [101]. In patients with PI3KCA WT, it is not associated with differences in these values [102]

Phosphatase and Tensin homolog (PTEN) is a negative regulator of the PI3K/AKT pathway, and mutations are seen in 5.8% of CRC patients [103,104]. Loss of this regulator results in permanent activation of the PI3K/AKT pathway, generating resistance to anti-EGFR drugs [103,104]. PTEN-negative status is correlated with worse response rates and lower PFS than PTEN-positive status [105,106].

### 2.3. BRAF Mutations

BRAF mutations are present in 5–21% of patients [107,108,109], mainly a valine to glutamic acid substitution at codon 600 (V600E), and are correlated with worse prognoses [110]. This mutation correlates with downstream MEK and ERK phosphorylation, making mitogen activation a permanently activated protein kinase (MAPK) pathway, promoting tumor cell growth and survival [111]. These mutations are associated with a high stage at diagnosis, poor differentiation, serrated architecture, and mucinous histology [111,112,113,114,115,116]. BRAFV600E-mutated CRC has a median survival of approximately one-third of non-BRAFV600E-mutated tumors (1 year vs. 2–3 years) [107,117]. The National Comprehensive Cancer Network (NCCN) recommends testing for mutations of KRAS/NRAS and BRAF genes in all patients with mCRC [118].

BRAF-inhibitor monotherapy in BRAFV600E-mutated CRC has low response rates, which is linked to incomplete inhibition of MAPK signaling in tumor cells [119,120]. BRAF inhibition generates a feedback activation of EGFR, causing an increase in the EGFR pathway and its consequences [121]. The addition of targeted therapies to BRAF and EGFR results in synergistic inhibition of BRAFV600E-mutant CRC [122,123]. Vemurafenib, the oral selective inhibitor of the BRAFV600 kinase, is suggested as a possibility for BRAFV600E mutation-positive patients by the NCCN [118], but studies vary in confirming its efficacy [124,125,126,127].

The phase III “Binimetinib, encorafenib, and cetuximab in BRAF600E-mutated CRC” (BEACON) trial showed the efficacy of another BRAF inhibitor, encorafenib, in dual-targeted EGFR and BRAF therapy as second-line systemic therapy for BRAFV600E-mutated CRC. In this study, the objective response rate (ORR) for the triple regimen (encorafenib and binimetinib, a MEK inhibitor, with cetuximab) was 26%, compared with 2% in the control group. The Median OS was 9 months for the triple regimen and 5.4 for the control group (*p* < 0.0001, 75) [123]. The use of triple therapy in clinical practice may be related to drug toxicity events, and drugs must be monitored constantly [128,129]. More recently, new local delivery technologies, nanocarriers (NCs), are being tested that propose to reduce these toxicity events by delivering medications precisely [129]. In the case of colorectal cancer, NCs that use CEA and CD44 receptors, both well-established biomarkers for CRC, have shown to be especially promising, with high specificity for tumor cells [130,131].

The Encorafenib, binimetinib and cetuximab in subjects with previously untreated BRAF-mutant CRC (ANCHOR-CRC) (NCT03693170) study released results in August 2022: the investigator-assessment confirmed that ORR based on local tumor assessments was 47.8% (95% CI 37.3–58.5%), and the median PFS per local review was 5.8 months (95% CI 4.6–6.4 months) [132].

Patients with mCRC who underwent chemotherapy with FOLFOX or CAPEOX in the last 12 months and are BRAFV600E-mutation-positive have as therapeutic options recommended by the NCCN the use of encorafenib plus cetuximab or vemurafenib [118].

### 2.4. HER2

HER2 is the only member of the EGFR family that is not activated by ligands but by homo or heterodimerization with other ligand-bound receptors [133], resulting in the activation of signal transduction pathways (RAS-RAF-ERK and PI3K-PTEN-AKT), which control cell growth and differentiation [134]. High HER2 expression is found in 2–11% of CRC cases [135]. HER2 amplification and KRAS, NRAS, and BRAF mutations simultaneously are very unlikely in advanced CRC [136].

In addition to functioning as an oncogenic driver, HER2 is a mediator of resistance to anti-EGFR therapies [137]. In patients with de novo HER2 amplification, PFS (PFS 89 vs. 149 days; *p* = 0.0013) and OS (307 × 515 days) are lower than in patients without amplification, when both treated with anti-EGFR targeted therapies [137].

Targeted therapy for the HER2 pathway in cancers in general currently relies on three medications with different mechanisms: trastuzumab, an HER2-targeted monoclonal antibody that binds to the extracellular domain; pertuzumab, a recombinant humanized monoclonal antibody that inhibits the heterodimerization of HER2 with other HER2 receptors; and lapatinib, the tyrosine kinase inhibitor against EGFR1 and HER2 [137].

The HERACLES-A, a multicenter open-label phase II trial (HER2 Amplification for Colorectal Enhanced Stratification), used trastuzumab plus lapatinib in HER2-positive KRAS exon 2 WT advanced CRC patients who were resistant to first-line treatments, including cetuximab [138]. In this study, an ORR of 30% was found with a median duration of response of 9.5 months and a median PFS of 21 weeks (95% CI 16–32 weeks) [138]. HERACLES-B, a phase II study with only twelve patients treated and eight assessed for response, evaluated the combination of pertuzumab and ado-trastuzumab emtansine (T-DM1), an antibody–drug conjugate linking trastuzumab to a microtubule inhibitor chemotherapy, in patients with mCRC HER2-positive KRAS exon 2 WT. In that study, a decrease in tumor volume was found in 87% of patients [139].

The MyPathWay (pertuzumab plus trastuzumab for HER2-amplified metastatic colorectal cancer NCT02091141) trial is still ongoing [140]. However, interim data point to a PFS of 2.9 months, an OS of 11.5 months, and an overall response rate gain of 32% in patients with mCRC HER2-amplified treatment with double treatment of trastuzumab and pertuzumab [141].

### 2.5. Immune Checkpoints

Mismatch-repair deficiency (dMMR) is found in 15% of all colorectal cancer patients [142]. Eighty percent of patients with dMMR are sporadic cases, primarily due to methylation of the mutL homolog 1 (MLH1) promoter gene. The hereditary cases are due to germline mutations in the MLH1 and mutS homolog 2 (MSH2) genes [143]. These events result in the cells being unable to repair mutations, causing a significant accumulation of these, generating tumors with high microsatellite instability (MSI-H) [144,145]. MSI-H is associated with increased risk for CRC, particularly with high tumor mutational burden and higher numbers of tumor-infiltrating lymphocytes [145,146]. The presence of dMMR alters the chance of responsiveness to specific conventional chemotherapy regimens and encourages the possibility of adding other treatments [147].

The immune system identifies the high tumor mutational burden generated by the propensity to mutations promoting a series of secondary signals that regulate the immune system response [148,149]. The programmed death 1/programmed death 1–ligand (PD-1/PD-L1) receptor-ligand system and Cytotoxic T lymphocyte-associated antigen 4 (CTLA-4) play a fundamental role in this regulation, being responsible for the tolerance or not of the immune system to those cells [150,151]. Malignancies often have elevated levels of checkpoint inhibitors such as CTLA-4 and PD-L1, significantly higher in mCRC, which interferes with the host immune response and confers resistance of these tissues to the host immune system [152]. Co-high expression of tumor PD-L1 and CTLA-4 in CRC tissues is a negative predictor of OS (HR 3.86, 95% CI 1.71–8.51, *p* = 0.001 in the membrane-bound receptor form and HR 5.72, 95% CI 1.87–14.54, *p* = 0.004 in the soluble form), in addition to being an independent prognostic factor for poor disease-free survival [153].

Immune checkpoint inhibitors (ICI) act on co-inhibitory receptors, such as CTLA-4 and PD-1, on immune system cells, or their ligands, such as PD-L1, on tumor and immune cells, in order to prevent the resistance of these cells to the immune system, potentiating the cytotoxic killing of tumor cells [154].

PD-1 blockage using antibodies in patients with MSI-H or dMMR mCRC has become an efficient treatment option [155]. Pembrolizumab, a humanized IgG4 antibody, has been approved for the treatment of mCRC by the FDA [156]. The recently published “Health-related quality of life in patients with MSI-high or dMMR mCRC treated with first-line pembrolizumab versus chemotherapy” (KEYNOTE-177) randomized, open-label, phase III study compared Pembrolizumab versus chemotherapy for MSI-H or dMMR in mCRC patients, with a median follow-up of 44.5 months (IQR 39.7–49.8). Although the study did not obtain a median overall survival for patients treated with Pembrolizumab due to the planned alpha of 0.025 for statistical significance not being achieved, data for PFS were promising [157]. The median PFS for the pembrolizumab-treated group was 16.5 months (95% CI 5.4–38.1) versus 8.2 months (6.1–10.2) for chemotherapy-treated patients (HR 0.59, 95% CI 0.45–0.79) [157]. A lower rate of serious adverse events was also found in patients treated with pembrolizumab (16%) compared with those treated with traditional chemotherapy (29%), pointing to pembrolizumab as an efficient option for first-line therapy in patients with MSI-H or dMMR mCRC [157].

Nivolumab, another humanized monoclonal IgG4-based PD-1 antibody, has been FDA-approved for use alone or in combination with ipilimumab, the CTLA-4 inhibitor, in patients with mCRC who have progressed following first-line chemotherapy treatment [118]. The study that resulted in this approval was the “Nivolumab in patients with metastatic DNA dMMR or MSI-high CRC” (CheckMate-142) study; the phase II, open-label, multicenter study first investigated the use of Nivolumab in patients with MSI-H or dMMR mCRC after the use of first-line therapy, finding an objective overall response rate of 31.1% (95% CI 20.8–42.9%), median PFS of 14.3 months (95% CI 4.3-NE), and 12-month OS of 73.4% (95% CI 62–82) [158]. In this same study, Nivolumab plus low-dose ipilimumab for MSI-H/dMMR was evaluated as a possible first-line treatment in mCRC patients [159]. ORR was 69% (95% CI 53–82), and the disease control rate was 84% (95% CI 70.5–93.5), with a 13% complete response rate. No median PFS or median OS was found with a follow-up of 24.2 months (24-month rates of 74 and 79%, respectively) [159].

Figure 1 and Table 1 summarize the described targeted therapies and important clinical trials cited throughout this publication.

## 3. Conclusions

Understanding the mechanisms involved in the pathophysiology and evolution of colorectal carcinoma, as well as possible specificities that can influence clinical presentation and outcomes of patients, allows for the development of more precise care. In this context, the identification of molecular subtypes, and the elaboration of strategies aimed at each of these, provide better final results for patients, fewer side effects, and reduce loss of therapeutic adherence.

## Figures and Tables

**Figure 1 cancers-15-03023-f001:**
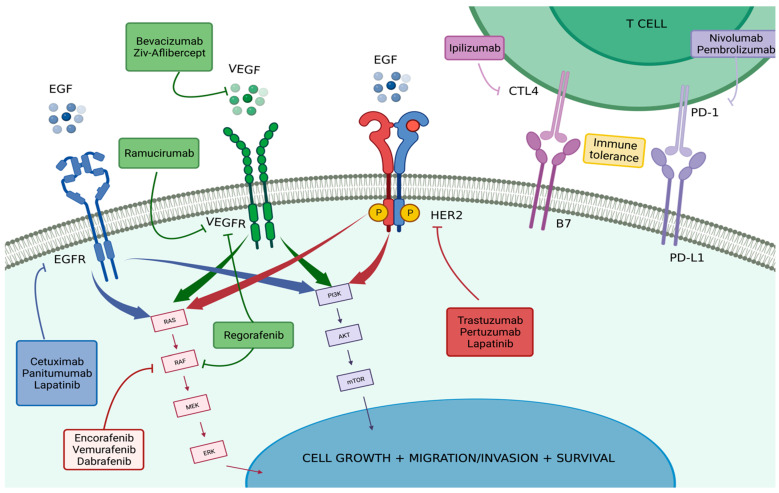
Targeted Therapies in Colorectal Carcinoma. Explores the mechanisms involved in the pathophysiology and development of colorectal carcinoma, drug interventions, and their targets within these mechanisms. The drugs mentioned are those approved by the FDA and already used in specific therapeutic strategies to treat CRC. All of these are explored more extensively throughout this review. Produced with biorender.com (accessed on 5 September 2022).

**Table 1 cancers-15-03023-t001:** Key Trials for Target Therapies. Summarizes the most commonly used target therapies in the treatment of mCRC by the NCCN guideline and some of its critical trials, exploring their specificities, such as characteristics, design, and results.

Target	Agent	Key trial	Characteristics	N	Design	Comparison	PFS	OS	RR
EGFR	Cetuximab	CRYSTAL NCT00154102	mCRC first-line	1198	Phase III, multi-center, randomized, parallel assignment, open label.	FX + Cetuximab vs. FX	8.9 vs. 8.0	19.9 vs. 18.6	46.9% vs. 38.7%
	Panitumumab	PRIME NCT00364013	mCRC first-line (KRAS WT)	1183	Phase III, multi-center, randomized, parallel assignment, open label	FX + Panitumumab vs. FX	9.6 vs. 8.0	23.9 vs. 19.7	55.21% vs. 47.68%
			mCRC first-line (KRAS Mutant)				7.3 vs. 8.8	15.5 vs. 19.3	39.53% vs. 40.2%
BRAF + MEK	Vemurafenib	SWOG NCT02164916	mCRC BRAFV600E-mutant	106	Phase II, multi center, randomized, crossover assignment, open label.	CIH + Vemurafenib vs. CIH	4.3 vs. 2.0	9.6 vs. 5.9	16% vs. 4%
	Dabrafenib + Trametinib	Concoran NCT01750918	mCRC BRAFV600E-mutant	43	Phase I, open-label	Dabrafenib	3.5	NR	12%
	Encorafenib + binimetinib	BEACON CRC NCT02928224	mCRC BRAFV600E-mutant	702	Phase III, multi-center, randomized, sequential assignment, open label.	Encorafenib + Binimetinib + Cetuximab vs. Irinotecan/cetuximab OR FOLFIRI/Cetuximab (Investigator’s choice)	4.3 vs. 1.51	9.03 vs. 5.42	29% vs. 2%
HER2	Trastuzumab + pertuzumab	MyPathway NCT02091141	mCRC HER2-amplified/ overexpressed	57	Phase IIA, multi-center, non-randomized, parallel assignment, open label.	Trastuzumab + pertuzumab	2.9	11.5	32%
	Trastuzumab + lapatinib	HERACLES NCT03225937	mCRC HER2 positive	54	Phase II, multi-center non-randomized, two sequential cohorts, open label.	Trastuumab + lapatinib	4.9	10.7	30%
VEGF	Bevacizumab	AVEX NCT00484939	mCRC elderly	280	Phase III, multi-center, randomized, parallel assignment, open label	Bevacizumab + capecitabine vs. Capecitabine	9.1 vs. 5.1	20.7 vs. 16.8	19% vs. 10%
	Regorafenib	CORRECT NCT01103323	mCRC refractory to all treatment	760	Phase III, multi-center, randomized, parallel assignment, quadruple masking.	Regorafenib + BSC vs. BSC	1.96 vs. 1.73	6.53 vs. 5.03	1% vs. 0.4%
	ZIV-aflibercept	VELOUR NCT00561470	mCRC refractory to oxaliplatin treatment	1226	Phase III, multi center, randomized, parallel assignment, triple masking	FOLFIRI + aflibercept vs. FOLFIRI + placebo	6.90 vs. 4.67	13.50 vs. 12.06	19.8% vs. 11.1%
	Ramucirumab	RAISE NCT01183780	mCRC refractory to all treatment	1072	Phase III, multi center, randomized, parallel assignment, quadruple masking	FOLFIRI + Ramucirumab vs. FOLFIRI + placebo	5.7 vs. 4.5	13.3 vs. 11.7	13.4% vs. 12.5%
PD-1	Pembrolizumab	KEYNOTE-164 NCT02460198	locally advanced unresectable CRC or mCRC + treatment refractory or dMMR/MSI-H	124	Phase II, multi center, non-randomized, single group assignment, open label	Pembrolizumab	2.3	31.4	32.8%
	Nivolumab	CheckMate142 NCT0206188	mCRC dMMR/MSI-H	74	Phase II, multi center, non-Randomized, parallel assignment, open label	Nivolumab	36% *	49% *	39%
PD-1 + CTLA-4	Nivolumab + Ipilimumab	CheckMate142 NCT0206188	mCRC dMMR/MSI-H	45	Phase II, multi center, non-Randomized, parallel assignment, open label	Nivolumab 3 mg/kg + 1 mg/kg Ipilimumab (4 doses) followed by Nivolumab 3 mg/kg	51% *	72% *	71%
				119		Nivolumab 3 mg/kg + Ipilimumab 1 mg/kg	54% *	71% *	65%

Abbreviations: MOA—mechanism of action PFS—Progression-free survival OS—Overall survival RR—Response rate mCRC—metastatic Colorectal Cancer FX—FOLFOX CIH—Cetuximab, Irinotecan Hydrochloride. BSC—Best Supportive Care *—48 months rate OS—survival RR—response rate MSI-H—Microsatellite instability-high PFS—Progression free survival PD-1—Programmed death-1 dMMR—deficient mismatch repair CTLA-4—Cytotoxic T lymphocyte-associated antigen 4.

## Data Availability

Data sharing not applicable.

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
