# Peer review of "Targeted Therapies in Colorectal Cancer: Recent Advances in Biomarkers, Landmark Trials, and Future Perspectives"

_cancers, 2023, doi:10.3390/cancers15113023_

Round 1

Reviewer 1 Report

The authors have written a well summarized review on the targeting approaches utilized so far for colorectal cancer therapy. However, there are some minor inclusions the author could do based on the comments below.

1.       There are some minor error in the caption of Table1 as it currently not readable and have overlapped with some other texts.

2.       Among many CRC based therapies, nanomedicine is one of the widely used approach for many cancers. The authors could site one such recent application of ‘cubosome nanoparticles tagged with Affimer proteins to target the CEA antigen on colorectal cancer in-vivo’.

3. Among targeted therapies authors could also highlight CD44 receptor which is well known biomarker for cancer cells including colorectal cancers. These CD44 receptors have been recently demonstrated to actively target drugs into colorectal cancer cells (https://doi.org/10.1021/acs.molpharmaceut.2c00439).

Author Response

Reviewer #1 Comments and Response: First, we would like to thank you enormously for the clear, detailed, and careful review. The comments were very accurate, both about opportunities for improvement and recommendations on how to achieve them. Your review was very important for the progress of the article. We hope that we have addressed each of your comments adequately. Thank you so much again.

  1. There are some minor error in the caption of Table 1 as it is currently not readable and have overlapped with some other texts.”

Thank you for the comment and observation. We have made changes to correct the errors and overlapped texts in Table 1.

  1. “Among many CRC based therapies, nanomedicine is one of the widely used approach for many cancers. The authors could site one such recent application of ‘cubosome nanoparticles tagged with Affimer proteins to target the CEA antigen on colorectal cancer in-vivo.”

Thank you very much for the excellent recommendation. The topic mentioned is, in fact, very current and very interesting. The indicated reference has been carefully studied and included in the new version of the Review. Considering the use of technology as a local delivery system and its gains in reducing side effects, the use of nanoparticles was included after discussing multiple target therapies and the possibility of increasing side effects, on Page 7, lines 503-512. Thank you again for the recommendation.

  1. “Among targeted therapies authors could also highlight CD44 receptor which is well known biomarker for cancer cells including colorectal cancers. These CD44 receptors have been recently demonstrated to actively target drugs into colorectal cancer cells (https://doi.org/10.1021/acs.molpharmaceut.2c00439).”

Thank you very much for the comment and the recommendation. As with CEA-targeted use, we studied and included the use of nanoparticles targeted by CD44 receptors in the study. The recommended source was essential for this. This addition, as well as the previous one, added enormously to the paper's quality gain and its current character.

Reviewer 2 Report

The authors have written an interesting paper and well presented manuscript. The topic is of great interest, as they review the actual biologyc biomarkers and targeted therapies used in the modern treatment of metastastatic colorectal cancer which has top interest.

The objective of the paper in well presented and described. The text is well organized and comprehensively. Also, the work is scientifically sound and nor misleading.

Table and Figure are illlustratives and informatives.

References are updated and completed.

The only remark I have is that authors do not explain how did they select the references included in the text and reviewed. Did they follow a pubmed research?. Which criteria did they follow. 

Author Response

Reviewer #2 Comments and Response:

  1. The authors have written an interesting paper and well presented manuscript. The topic is of great interest, as they review the actual biologyc biomarkers and targeted therapies used in the modern treatment of metastastatic colorectal cancer which has top interest. The objective of the paper in well presented and described. The text is well organized and comprehensively. Also, the work is scientifically sound and nor misleading. Table and Figure are illlustratives and informatives. References are updated and completed.

Thank you for all the positive comments. We greatly appreciate the thoughtful individual assessment carried out. It is of immense importance for us and the quality of the production of scientific knowledge in general, as proposed by this journal, the revision and validation of the contents created by references in the area like you.

  1. The only remark I have is that authors do not explain how did they select the references included in the text and reviewed. Did they follow a pubmed research?. Which criteria did they follow. 

Thank you very much for your comment and observation. The included references were selected through Pubmed, Embase, and Cochrane searches using the keywords "Colorectal Cancer," "Biomarkers," and "targeted therapies," in addition to individualized searches of each of the biomarkers and medications that proved to be relevant throughout the extensive bibliographical research. Following the pattern of other reviews in the journal Cancers, the review was not carried out systematically since it is intended to be a broader literature review. Thus, a detailed and rigorous explanation of the methodology was not included in the paper. We enclose below the information of all the reviews of this genre included in the last issue of Cancers:

https://doi.org/10.3390/cancers15112934

https://doi.org/10.3390/cancers15112933

https://doi.org/10.3390/cancers15112932

https://doi.org/10.3390/cancers15112929
